# Immunopathologic Role of Eosinophils in Eosinophilic Chronic Rhinosinusitis

**DOI:** 10.3390/ijms232113313

**Published:** 2022-11-01

**Authors:** Seung-Heon Shin, Mi-Kyung Ye, Jinwoo Park, Sang-Yen Geum

**Affiliations:** Department of Otolaryngology-Head and Neck Surgery, School of Medicine, Catholic University of Daegu, Daegu 42472, Korea

**Keywords:** eosinophils, chronic rhinosinusitis, extracellular trap, *Staphylococcus aureus*, fungus

## Abstract

Chronic rhinosinusitis (CRS) is a diverse chronic inflammatory disease of the sinonasal mucosa. CRS manifests itself in a variety of clinical and immunologic patterns. The histological hallmark of eosinophilic CRS (ECRS) is eosinophil infiltration. ECRS is associated with severe disease severity, increased comorbidity, and a higher recurrence rate, as well as thick mucus production. Eosinophils play an important role in these ECRS clinical characteristics. Eosinophils are multipotential effector cells that contribute to host defense against nonphagocytable pathogens, as well as allergic and nonallergic inflammatory diseases. Eosinophils interact with *Staphylococcus aureus*, Staphylococcal enterotoxin B, and fungi, all of which were found in the tissue of CRS patients. These interactions activate Th2 immune responses in the sinonasal mucosa and exacerbate local inflammation. Activated eosinophils were discovered not only in the tissue but also in the sinonasal cavity secretion. Eosinophil extracellular traps (EETs) are extracellular microbes trapping and killing structures found in the secretions of CRS patients with intact granule protein and filamentous chromatic structures. At the same time, EET has a negative effect by causing an epithelial barrier defect. Eosinophils also influence the local tissue microenvironment by exchanging signals with other immune cells and structural cells. As a result, eosinophils are multifaceted leukocytes that contribute to various physiologic and pathologic processes of the upper respiratory mucosal immune system. The goal of this review is to summarize recent research on the immunopathologic properties and immunologic role of eosinophils in CRS.

## 1. Introduction

Chronic rhinosinusitis (CRS) is a common chronic disease that has an impact on the socioeconomic burden, quality of life, and healthcare costs [1,2,3]. Although the etiology and pathogenesis are not completely understood, noninfectious inflammation plays a key role and is characterized by chronic inflammation of the sinonasal mucosa with inflammatory infiltration of lymphocytes, monocytes, plasma cells, and eosinophils, as well as pathologic change of mucoepithelial structures [4,5]. CRS is clinically, pathologically, and immunologically diverse. It can classify patients based on clinical characteristics, such as phenotypic classification, nasal polyp presence, atopic status, recurrent disease, severity, presence of complications, and concomitant diseases. Based on the endotype dominance, CRS is divided into either type 2 or non-type 2 [6,7,8]. Non-type 2 CRS can be divided into type 1 and type 3. Type 1 inflammation is distinguished by increased interleukin (IL)-12 and interferon (IFN)-γ, as well as CD8+ T cells in sinonasal tissue with purulent nasal discharge. Type 2 is distinguished by increased levels of IL-4, IL-5, IL-13, and eosinophils in sinonasal tissue, as well as olfactory dysfunction in the early stage. Type 3 inflammation is characterized by increased levels of IL-17, IL-22, and neutrophils in sinonasal tissue. Based on the clinical phenotypes, diffuse CRS is divided into eosinophilic CRS (ECRS) and non-eosinophilic CRS (NECRS) [7].

The majority of western CRS (especially CRS with nasal polyps (NP)) is classified as type 2 inflammatory endotype (73–87%) and about 17–61% in Asian CRS [9,10]. Eosinophilic infiltration is a hallmark of type 2 inflammation. In white people, eosinophilic inflammation is present in 65–90% of NP cases, whereas it is present in less than 50% of Asian people. Type 2 or ECRS has recently become more common in Asian regions [10,11]. ECRS and NECRS have differences in their immunopathologic characteristics. ECRS has a higher recurrence rate, more frequent comorbid asthma, and severe clinical symptoms compared to NECRS [12,13]. Although eosinophilia is a hallmark of type 2 CRS, neutrophilic inflammation is also common in the tissue of severe type 2 CRS [14]. The presence of a Charcot-Leyden crystal, which is common in ECRS, is strongly linked to neutrophil infiltration [14,15]. The production of chemokines by epithelial cells in response to the Charcot–Leyden crystal induces neutrophil migration and activation. Activated neutrophils can promote eosinophil transmigration and eosinophil accumulation. In ECRS, the presence of both eosinophils and neutrophils is associated with disease severity and a difficult-to-treat phenotype. Patients with mixed granulocyte infiltration and an increased inflammatory burden have a worse computed tomographic score, poor olfactory function, a lower quality of life, and poor treatment outcomes [16]. Neutrophils may also play a role in type 2 inflammation.

Eosinophils were previously thought to be end-stage effector cells involved in parasite infections and hypersensitivity diseases, releasing granule proteins, lipid mediators, and other inflammatory molecules. Recent research has shown that eosinophils, which are multifunctional leukocytes with immunoregulatory functions, can be found not only in inflammatory diseases but also in the absence of pathologic conditions [17,18]. At baseline conditions, eosinophils maintain the host’s immune homeostasis. Eosinophils have an autocrine and paracrine regulatory loop that promotes tissue eosinophil recruitment and survival through the production and interaction with cytokines, chemokines, complements, and growth factors [17]. Transforming growth factors (TGFs), vascular endothelial growth factor, and metalloproteinase affect tissue cells; for example, TGF-β1 induces epithelial–mesenchymal transforming, stromal fibrosis, and basement membrane thickening in nasal mucosa [19,20]. Innate and adaptive immune responses against pathologic organisms can activate eosinophils. Activated eosinophils defend against large, nonphagocytable organisms. Eosinophils have antibacterial activity as well, as they release mitochondrial DNA in response to bacteria [21]. When eosinophils interact with commensal bacteria or fungi, they have the potential to initiate or exacerbate local inflammatory responses. Moreover, during host defense, eosinophils produce a variety of chemical mediators that can induce detrimental effects on the host. Elevated blood eosinophil counts correlated with tissue eosinophilic inflammation and to the disease severity and the recurrence of CRS after sinus surgery [22,23]. Eosinophil counts in peripheral blood might be a biomarker for severe intractable CRS with type 2 inflammation. Although tissue eosinophilia with prolonged survival and activation within the sinonasal mucosa is a common pathologic feature of type 2 CRS, the immunopathologic role of eosinophils in CRS is still incompletely defined. We present here a summary of the biological properties and role of eosinophils in type 2 ECRS.

## 2. Clinical and Immunological Features of ECRS

ECRS is defined as refractory or intractable CRS with profound eosinophil infiltration and a strong proclivity for recurrence after surgery. ECRS was previously classified into four subtypes based on underlying etiologies and therapeutic interventions, including (1) superantigen-induced ECRS, (2) allergic fungal sinusitis, (3) nonallergic fungal eosinophilic sinusitis, and (4) aspirin-exacerbated ECRS [24]. The most severe form of ECRS is aspirin-exacerbated ECRS, which is caused by a combination of aspirin sensitivity, asthma, and NP. A variety of stimuli and mechanisms for the development of ECRS have been proposed. However, other mechanisms and categories may exist.

Most ECRS develop in adults with high viscous nasal secretion with olfactory dysfunction precedes other nasal symptoms with multiple bilateral NP. In earlier stages of disease, computed tomogram (CT) images show ethmoid sinus dominant involvement with opacification of the posterior ethmoid sinus and the olfactory cleft [25,26]. Asthma or airway hyper-responsiveness are frequently associated with ECRS. ECRS is a refractory chronic inflammatory disease with a higher recurrence rate after sinus surgery, and post-operative therapy with nasal irrigation, systemic or intranasal corticosteroid, or both is commonly used to control ECRS [25,26]. Although there were no definite criteria for eosinophilia until now, ECRS patients had a higher peripheral blood eosinophil absolute count and percentage, and patients with more than 10% eosinophils in peripheral blood had a significantly higher recurrence or refractoriness [27,28]. Because the peripheral blood eosinophil counts significantly correlate with tissue eosinophils, peripheral blood eosinophilia is of diagnostic value for ECRS. Researchers and studies have different diagnostic criteria for eosinophilic infiltration in ECRS [27,29]. A large-scale multicenter epidemiological study in Japan proposed a diagnostic guideline for ECRS [27]. In this study, ECRS is defined as a score of 11 or higher for the sum of the four factors (Table 1). They also proposed a diagnostic cut-off value of diagnostic criteria for tissue eosinophilia if the mean number of three fields of tissue eosinophil count is 70 or higher by microscopic examination in a high-power field.

Increased eosinophilopoiesis in bone marrow, increased migration of eosinophils to the target organ, and increased eosinophils survival are required for eosinophils accumulation in blood and tissues. IL-3, IL-5, and GM-CSF are important hematopoietic factors for eosinophil development, differentiation, and maturation. Eosinophils’ survival is prolonged by IL-3, IL-5, GM-CSF, and TNF-α [18]. Interaction of cell surface adhesion molecules, such as E-selectin, L-selectin, P-selectin, and β1-integrin as well as vascular cell adhesion molecule-1, and intercellular adhesion molecule-1 play an important role in eosinophil migration from the bloodstream into the tissues [29]. Tissue eosinophilia develops in patients with systemic atopic disease who have high total serum IgE and Th2 cytokines. Patients with atopic CRS have a significantly higher prevalence of tissue eosinophilia [30]. Although CRS can develop regardless of the host’s atopic status, atopy increases the risk of developing CRS. In 25–70% of cases, both diseases can coexist in the same patient [11,31]. Some studies, however, have found that systemic atopy has no effect on tissue eosinophilia in CRS, and in the case of ECRS, the complication of atopic diathesis is not necessary [32,33]. Although atopy with systemic eosinophilia is not the primary cause of tissue eosinophilia, an increase in peripheral blood eosinophils can induce nonspecific eosinophil recruitment into sinonasal mucosa and is associated with tissue eosinophil counts [34]. As a result, atopy may account for some of the tissue eosinophilia in CRS. In contrast to systemic atopic status, approximately 80% of CRSwNP tissues in Western countries have highly increased local IgE, eosinophils, and Th2 cytokines [35]. Th2 cytokines, such as IL-4, IL-5, and IL-13, directly or indirectly promote eosinophilia by influencing eosinophil differentiation, survival, and activation. These Th2 cytokines have been suggested as potential predictors of clinical severity and treatment outcome. Among several Th2 cytokines, tissue and nasal secretion IL-5 seems to be an important biomarker, which correlated with tissue eosinophil count and clinical severity of CRS [36,37].

Proinflammatory cytokines; TNF-α and IFN-γ, produced from endothelial cells and fibroblasts in sinonasal mucosa induce the migration of eosinophils from blood [38]. Epithelial cell-derived cytokines, such as IL-25, IL-33, and thymic stromal lymphopoietin (TSLP) initiate type 2 inflammation by activating local type 2 immunity and increasing Th2 cytokine production. IL-25 promotes eosinophil infiltration increasing Th2 cytokine production from memory Th2 cells, basophils, and group 2 innate lymphoid cells [39]. IL-33 promotes innate type 2 inflammation, whereas TSLP promotes both innate and adaptive type 2 inflammation [40,41]. These cytokines produced by epithelial cells are significantly elevated in ECRS [42].

## 3. Clinicopathologic Role of Eosinophils in ECRS

Early development of olfactory dysfunction with opacification of the posterior ethmoid sinus and the olfactory cleft in CT images characterizes ECRS. Olfactory dysfunction in ECRS is associated with more bilateral lesions, and a more severe disease status, involving the olfactory epithelium around the ethmoid and olfactory cleft areas. Olfactory dysfunction is associated with odorant passage into the olfactory epithelium, which may be hampered by diseased mucosal inflammation [43]. The level of inflammatory cytokines and the degree of eosinophil infiltration in the sinonasal mucosa influence the development of olfactory dysfunction [44]. Although tissue eosinophilia is associated with olfactory dysfunction with impaired odor threshold score, the increased number of eosinophils alone, however, had no effect on olfactory function in CRS patients [45]. Eosinophil-derived granule proteins or eosinophil-associated cytokines are both neurotrophic and neurotoxic; in ECRS, they may destroy olfactory epithelium and affect the survival of olfactory neural epithelial cells. Th2 and proinflammatory cytokines, as well as Th1 cytokines, influence the development of olfactory dysfunction in CRS [44,46]. To elucidate the pathophysiologic mechanism of olfactory dysfunction in CRS, biologic markers, such as eosinophil-derived granule protein levels, tissue cytokine levels, and other inflammatory markers must be determined.

Mucus hypersecretion is one of the major symptoms of upper and lower airway inflammatory diseases. Goblet cell metaplasia in association with mucus hypersecretion is an important pathologic finding in CRS, and MUC4, MUC5AC, MUC5B, and MUC8 are commonly upregulated [47]. Eosinophils enhance the expression of MUC4, MUC5B, and MUC8 mRNA in rhinovirus-infected nasal epithelial cells through the interaction of intracellular adhesion molecule-1 and the ligands of eosinophils [48]. Adhesion of epithelial cells to activated epithelial cells could be involved in enhancing eosinophil degranulation [49]. Eosinophil–epithelial interactions enhance the secretion of MUC5AC mucin and profibrotic cytokines involving tissue remodeling and secreted eosinophil products induce mucus production from airway epithelial cells [50].

Tissue remodeling is a complex process by the interactions among inflammatory cells such as eosinophils, neutrophils, and lymphocytes with structural epithelial cells and fibroblasts. Eosinophil infiltration in CRS is associated with the integrated process of nasal epithelial cells, extracellular matrix (ECM), inflammatory cells, and their chemical mediators [51]. Activated eosinophils induce airway mucosa stromal fibrosis, epithelia damage, increased interstitial edema, and increased production of ECM by extracellular deposition of eosinophil granule proteins and inflammatory mediators [52]. Eosinophils are a source of inflammatory molecules implicated in tissue remodeling by the production of TGF-α, TGF-β1, fibroblast growth factor, vascular endothelial growth factor, IL-13, and IL-17. These molecules induce dysregulation of ECM homeostasis with tissue remodeling and fibrosis. Physical contact between activated eosinophils and fibroblasts enhances ECM mRNA, such as collagen type 1, tissue inhibitors of matrix metalloproteinase, matrix metalloproteinase-9, and α-smooth muscle actin [53].

## 4. Eosinophils, *Staphylococcus aureus*, and Staphylococcal Enterotoxin B (SEB)

One of the most common causes of bacterial CRS is *S. aureus*. *S. aureus* is colonized in the sinonasal mucosa and has been linked to chronic inflammatory airway diseases, such as allergic rhinitis, allergic asthma, and CRS. Staphylococcus colony count was much higher in CRSwNP than in control, and SEs are the classical causative allergen of CRS [54]. *S. aureus* can activate Th2 immune responses by not only enterotoxins but also microbial pathogenic components. *S. aureus* exacerbates type 2 chronic airway inflammatory diseases by directly inducing the production of epithelial cell-derived cytokines via the toll-like receptor 2 and nuclear factor-κB pathway, as well as inducing the production of Th2 cytokines, IL-5 and IL-13 from sinonasal tissue. SEB can cause Th2 cytokine production and eosinophilic infiltration in CRS patients [55,56]. Type 2 immune responses in airway mucosa are induced by *S. aureus* serine protease-like proteins found in nasal polyp tissues. They increase Th2 cytokine production and eosinophil infiltration of the airways [57,58].

SEB was found in patients with different subtypes of CRS, and these SEs act as disease modifiers and superantigens in the sinonasal mucosa [8,59]. When SEB binds to MHC class II molecules or T-cell receptor β chain, the immune system becomes overly activated, resulting in polyclonal T-cell activation. CRS sinonasal mucosa has significantly higher levels of specific IgE against SEs [60]. The ECRS tissue showed increased eosinophil infiltration, as well as higher SEB and SE-specific IgE levels. SEB is associated with not only localized eosinophilic inflammation but also the release of eosinophilic granule proteins with eosinophil activation. The level of SE-IgE was associated with disease severity, eosinophilic inflammation, level of cationic protein, and an increased incidence of asthma comorbidity in ECRS patients. SEB increased Th2-deviated CD4+ T-cell expansion in patients with CRS, which was associated with increased tissue inflammation and was related to disease severity [60]. Intranasal administration of SEB aggravates airway inflammation by increasing eosinophil infiltration and the production of proinflammatory cytokines by a variety of immune cells [55,61]. SEB exposure increases the number of eosinophils in bone marrow, circulating eosinophils, and finally induces tissue eosinophilia with elevated levels of Th2 cytokines, exotoxin, and increased expression of CCR3 and adhesion molecules [61]. SEB causes tissue damage and remodeling by inhibiting regulatory T cells, increasing Th2 cytokine production, and enhancing eosinophil and mast cell functions [57]. SEB can also impair nasal epithelial cells’ innate physical barrier defense system. When SEB was applied to nasal epithelial cells, the expression of zonula occludens-1 and occluding protein was decreased, which was associated with increased cellular permeability and decreased trans-epithelial electrical resistance [62]. Simultaneously, SEB-induced IL-6 and IL-8 production from epithelial cells, as well as endoplasmic reticulum stress, can cause nasal epithelial cell barrier dysfunction.

## 5. Eosinophils and Fungi

Airborne fungi are ubiquitous and are deposited on the airway mucosa upon inhalation. The components of the fungal cell wall act as pathogen-associated molecular patterns that are recognized by pattern recognition receptors of the host innate immune system, such as toll-like receptors and C-type lectin receptor. They stimulate cytokine and chemokine production, immune cells recruitment, and antimicrobial responses. Th cells play an important role in the adaptive immune response to fungi. The Th1 response acts as a protector, whereas the Th2 acts as a non-protective role. The balance between the protective inflammatory response of Th1 cells and the pathologic inflammatory response of Th2 cells is controlled by regulatory T cells [63]. Fungi are frequently discovered in patients with type 2 airway diseases who have elevated levels of local and systemic eosinophils. Fungi, according to some researchers, may play a role in the pathogenesis of CRS via interaction with epithelial cells and the innate immune system of the airway mucosa [48,64]. Fungal protease can facilitate pathogen access to the target tissue by inducing epithelial barrier dysfunction and altering junctional complex protein expression [65]. The loss of epithelial barrier integrity is commonly found in fungi-mediated type 2 chronic inflammatory airway diseases. Fungi-induced production of eosinophilia-associated cytokines, such as IL-5 and IL-13, from nasal mucosa and epithelial cell-derived cytokines, the initiating mediators of type 2 immune responses. Fungi-induced increases in the production of epithelial cell-derived cytokines, the expansion of ILC2s, tissue eosinophil infiltration, and Th2 lymphocytic cell recruitment and proliferation.

Eosinophils are important immune cells that protect the host against pathogenic fungi by releasing cytotoxic granules, such as major basic protein, eosinophil-derived neurotoxin, eosinophil cationic protein, and eosinophil peroxidase. Eosinophils interact with fungal cell wall components, such as β-glucan, chitin, and proteases. Fungi can directly activate or induce the release of granule proteins in several ways: (1) interaction with G protein-coupled receptor in a calcium-dependent manner [66], (2) interaction of fungal β-glucan with β2 integrin molecule of cell membrane [67], (3) interaction of fungal aspartate protease with protease activated receptor-2 [68], and (4) interaction of fungal hyphae or conidia with the CD11b extracellular I-domain [69]. Fungi can also indirectly induce the accumulation of eosinophils through the enhanced production of IL-33 or cysteinyl leukotrienes in airway mucosa [70] These chemical mediators activate IL-5 and IL-13 producing group 2 innate lymphoid cells, and then induce the production, recruitment, and activation of eosinophils.

Eosinophils and their degranulation products are abundant in the eosinophilic mucin of CRS patients. Eosinophils cluster in the mucus around fungal elements, implying that eosinophils target fungi in the mucus of CRS patients and release toxic protein granules to remove fungi [71]. Fungal cell wall protease, chitin, and β-glucans initiate fungicidal activity of eosinophils by activating and releasing their granule proteins [69]. In eosinophil-deficient mice, intratracheal instillation of *Aspergillus fumigatus* resulted in impaired fungal clearance and increased fungal germination. These antifungal activities of eosinophils are not associated with direct contact with fungal components, but rather with fungi producing proinflammatory cytokines and chemokines [72].

## 6. Eosinophil Extracellular Trap (EET)

Extracellular trap cell death is a type of rapid cell death distinct from apoptosis and necrosis, characterized by the release of intact cytoplasmic organelles via nuclear and plasma membrane breakdown [73]. Extracellular traps are produced by neutrophils and eosinophils. After stimulation with immobilized immunoglobulins, calcium ionophore, platelet-activating factor, or phorbol myristate acetate, eosinophils can secrete filamentous nuclear DNA and granule contents in a nicotinamide adenine dinucleotide phosphate oxidase-dependent mechanism and lipopolysaccharide, C5a, and CCL11 induce noncytolytically release of mitochondrial DNA [74,75]. EET can appear in infectious, autoimmune, and noninfectious diseases. EET is most commonly found in inflamed tissue or luminal spaces, such as the airway or gastrointestinal mucosa. EETs are found at the site of airway damage and protect against the invasion of external pathogens in CRS [76]. EET with tissue eosinophilia was found in significantly higher concentrations in ECRS patients than in non-ECRS patients. EET formation is triggered by IL-5 and TSLP, both of which are commonly elevated type 2 cytokines in ECRS. EETs were found in inflamed nasal mucosa with eosinophil infiltration in both ECRS and non-ECRS patients [77]. These findings may be related to CRS’s histologic heterogeneous disease characteristics. However, the pathophysiologic mechanism underlying the formation of EET in non-ECRS is not well understood. Hwang et al. [77] proposed that non-ECRS tissues exhibit a wide range of cytokines with Th1, Th2, and Th17-type inflammation patterns. These type 2 cytokines may also play a role in the development of EET in non-ECRS patients. Increased eosinophils and EETs were found in the subepithelial layer, particularly near epithelial defects. The major extracellular structural components of ECRS secretion are intact eosinophil granules and nuclear-derived DNA [78].

EET is an innate immune defense mechanism that traps and destroys extracellular bacteria, fungi, and viruses [76]. Eosinophils produce EET through both direct and indirect interactions with *S. aureus*. Direct contact between *S. aureus* and the eosinophil is likely to initiate the formation of EET. An in vitro study found that eosinophils migrate toward *S. aureus* and entrap both inside and outside of diseased nasal mucosal tissue, producing a large amount of EET [78]. *S. aureus* attracts eosinophils and induces EET to remove or inhibit the growth of bacteria by enzymatic digestion of extracellular DNA with granule proteins. In general, the most abundant proteins in nuclear-derived extracellular traps are histones that have cytotoxic effects [79]. In ECRS, EET was associated with Th2 inflammation, and *S. aureus* induced EET through the reactive oxygen species (ROS) dependent pathway [78]. Eosinophils primed with cytokine release mitochondrial DNA and granule proteins with antimicrobial properties in the extracellular space. Because eosinophil DNA traps do not have antibacterial properties, antibacterial properties must be demonstrated by both intact DNA and enzymatic granule proteins.

The histological hallmark of ECRS, particularly allergic fungal rhinosinusitis or eosinophilic mucin rhinosinusitis is eosinophilic mucin, a thick and glue-like secretion. These thick mucus secretions are associated with both local inflammations-induced mucin overproduction and eosinophil-induced mucin glycoprotein production from nasal epithelial cells. Immunostaining of eosinophilic mucin revelated that they contain increased contents of immune cells, Charcot–Leyden crystals, and large polymers including nuclear-derived chromatin structures [76]. EET is associated with disease severity and affects the viscosity of eosinophilic mucin in ECRS patients [77]. Fungal conidia and eosinophils provide an adhesive surface for microorganism entrapment and EET formation. *Aspergillus fumigatus* causes the release of EET with toxic granule proteins in the extracellular matrix, which has fungicidal and fungistatic properties [80]. *Aspergillus fumigatus* induces the release of EETs in an NADPH oxidase- and mitochondrial ROS-independent manner, whereas neutrophil extracellular DNA trap development is NADPH oxidase-derived ROS dependent [80].

Because EET contains toxic granule proteins, it can both benefit and harm the sinonasal mucosal immune defense system. If the human immune system recognizes bacteria or fungi as a pathogen in CRS patients, eosinophils may attack them by recruiting and activating in the airway lumen. Although EETs have host protective functions, they can also aggravate inflammation in eosinophilic airway diseases by damaging the surrounding tissues and developing epithelial barrier dysfunction [76,78]. EET, on the other hand, may play a critical role in maintaining barrier function after inflammation-associated epithelial cell damage, thereby protecting the host from persistent pathogenic invasion.

## 7. Eosinophils and Local Tissue Immune Cells

By exchanging signals, eosinophils interact with leukocytes, lymphocytes, and resident tissue cells to maintain the local immune microenvironment (Figure 1). Chemical mediators produced by T cells initiate eosinophil recruitment to the site of inflammation, while eosinophils promote T-lymphocyte chemotaxis via chemokine expression and modulate local Th2 inflammation immune responses via Th2 cytokine production. Eosinophils also promote T-cell recruitment by secreting chemokines from respiratory epithelial cells [81]. Although eosinophils are the most prominent modulators of Th2 immune responses, they can also control Th1 inflammatory responses. Eosinophils produce IL-12, IFN-γ, and toll-like receptors, all of which are associated with Th1 immune responses [82]. As a result, eosinophils may play a role in regulating the balance of Th1-Th2 immune responses. Eosinophils may dampen local inflammatory responses by producing IL-10 and TGF-β, which modulate T-regulatory cell activity [17,82]. Eosinophils also interact with mast cells and basophils, which increase eosinophil recruitment, survival, activation, and proliferation, which then significantly modulates Th2 immune responses [17].

The interaction of eosinophils with airway epithelial cells is a critical mechanism for initiating local tissue inflammation, as evidenced by increased production of proinflammatory cytokines, chemokines, and cell surface adhesion molecules, such as ICAM and VCAM-1 [83]. These adhesion molecules are involved in granule protein release, chemotaxis, and migration of eosinophils. The interaction of eosinophils and epithelial cells is mediated by IFN-γ, TNF-α, IL-1, and several chemokines. Increased tissue eosinophils and their activation by epithelial cell-derived cytokines and Th2 cytokines, eosinophils contribute to the development of tissue remodeling by releasing cytotoxic granules that facilitate and initiate epithelial cell damage. TGF-β production by the epithelial cells and eosinophils contributes to extracellular matrix accumulation and the development of epithelial–mesenchymal transition with mesenchymal cell proliferation [84]. Activated eosinophils can also cause the production of mucin from airway epithelial cells [48]. As a result, the interaction of eosinophils and epithelial cells is critical for the initiation, development, aggravation, and resolution of airway inflammation.

In CRS, eosinophils and fibroblasts may be a source of profibrotic cytokines for tissue remodeling. Activated eosinophils cause fibroblasts to become activated and differentiated into myofibroblasts. TGF-α, TGF-β1, fibroblast growth factor, vascular endothelial growth factor, IL-13, and IL-17 are eosinophil fibrogenic mediators that induce a fibrogenic phenotype in fibroblasts and dysregulation of extracellular matrix homeostasis, ultimately leading to tissue remodeling and fibrosis [51]. The cellular interaction of eosinophils and fibroblasts is critical not only for local immune responses but also for tissue remodeling and repair.

## 8. Summary and Conclusions

CRS is defined as chronic inflammation of the sinonasal mucosa characterized by elevated cytokines, chemokines, and lipid mediators, as well as inflammatory cell infiltration. Despite the presence of neutrophil infiltration in severe type 2 CRS, increased eosinophilic inflammation is a hallmark of type 2 inflammation [14]. Although the pathophysiologic mechanisms of tissue eosinophilia in CRS are not completely understood, Th2 cytokines, eosinophil-selective chemokines, and EET may be associated with eosinophil accumulation. Eosinophils are recruited from bone marrow to blood and the site of immune responses during type 2 inflammation. These eosinophils have been identified as important effector immune cells that protect the host from the invasion of pathogens, such as parasites, fungi, bacteria, and viruses [85]. In CRS, *S. aureus*, SEB, and fungi induce type 2 inflammation with eosinophil infiltration in tissue and sinonasal mucosa. The coexistence of fungi and SEB synergistically stimulate type 2 immune responses. These immune responses may be related to destructive leukocytic activities by granule release as a host defense mechanism against nonphagocytable pathogens. EETs are found in the secretions of ECRS patients. Eosinophil DNA traps extracellular microorganisms that can entrap intact granule proteins. These free toxic granule proteins destroy not only pathogenic organisms but also the function of the epithelial barrier. As a result, eosinophils have multifaceted immunobiological characteristics that can be beneficial or harmful in the development of CRS. Eosinophils have immunomodulatory properties because they interact with a variety of immune and structural cells. In the development of CRS, especially ECRS, eosinophils contribute as significant effector cells and the eosinophils remain as a harmful immune cell exacerbating local inflammatory responses, resulting in poor therapeutic results, and a high prevalence of comorbid diseases. For the treatment of severe, uncontrollable ECRS, several biologics targeting the underlying type 2 inflammation are developed and approved, such as dupilumab, omalizumab, and mepolizumab [86]. Dupilumab (anti-IL-4Rα monoclonal antibody) and mepolizumab (anti-IL-5 monoclonal antibody) more directly affect eosinophils. Although omalizumab (anti-IgE monoclonal antibody) does not directly influence eosinophils, there is a significant correlation between levels of total IgE and levels of IL-5, ECP, and number of eosinophils [34]. These biologicals can improve the sense of smell, quality of life, nasal polyp scores, and CT score, reducing the need for rescue surgery with somewhat different efficacy and safety [87]. However, treatment with dupilumab can cause transient blood eosinophilia, which is related to the inhibition of eosinophil migration to the tissue [88,89]. Although most of the eosinophilia is transient and does not develop clinical symptoms, treatment with prednisolone or dual therapy with anti-IL-5/5R monoclonal antibody may be needed in some cases [89]. Therefore, the selection of biologics should be personalized depending on the pathophysiology of CRS. If can gain precise knowledge about the immunopathologic role of eosinophils in the pathogenesis of CRS, controlling eosinophil maturation, recruitment, survival, and activation could be effective therapeutic strategies.

## Figures and Tables

**Figure 1 ijms-23-13313-f001:**
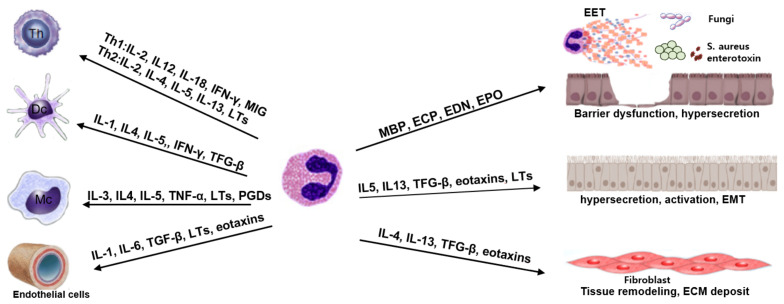
Eosinophil mediators and the interaction of eosinophils with tissue microenvironments. Th: Th cells, Dc: dendritic cells, Mc: mast cells, EET: eosinophil extracellular trap, EMT: epithelial–mesenchymal transition.

**Table 1 ijms-23-13313-t001:** Diagnostic criteria of eosinophilic chronic rhinosinusitis (Adapted from Ref. [27]).

Factor	Score
Disease side: both sides	3
Nasal polyp	2
CT shadow: ethmoid ≤ maxillary	2
Eosinophils of peripheral blood	
2< and ≤5%	4
2< and ≤10%	8
10<	10

## Data Availability

Not applicable.

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
