# Peer review of "Immunopathologic Role of Eosinophils in Eosinophilic Chronic Rhinosinusitis"

_ijms, 2022, doi:10.3390/ijms232113313_

Round 1

Reviewer 1 Report (Previous Reviewer 2)

The authors have now completed work that can be published. The chapter is a helpful summary of eCRS and what role eosinophils may play.

Please read and correct typos/spelling errors 

Line 12

line 46

In line 303, the line on DNA is not clear. Just explain more clearly.

Line 44, do you mean ' based on '' endotypes''...not phenotypes

Author Response

I thank the editors and referees of the ‘International Journal of Molecular Sciences’ for taking their time to review my article.

I have made some corrections and changed some parts in the manuscript after going over the referee’s comments.

Please read and correct typos/spelling errors 

Line 12

Answer) Corrected spelling error: nonphagocytable

line 46

In line 303, the line on DNA is not clear. Just explain more clearly.

Answer) DNA was changed as ‘eosinophil DNA traps’

Line 44, do you mean ' based on '' endotypes''...not phenotypes

Answer) Based on the EPOS 2020 and Contemporary Classification of Chronic Rhinosinusitis Beyond Polyps vs No Polyps: A Review JAMA Otolaryngol Head Neck Surg. 2020 Sep 1;146(9):831-838.

‘Based on the clinical phenotypes, diffuse CRS is divided into eosinophilic CRS (ECRS) and non-eosinophilic CRS (NECRS)’

I hope the revised manuscript will better meet the requirements of the ‘International Journal of Molecular Sciences’ for publication.

Thank you.

Reviewer 2 Report (New Reviewer)

This review is very interesting and worth in the field of Otolaryngology. This article is out of scope of this journal. So, you should submit any other journal in the field of Otolaryngology.

Author Response

Thank you for kind comments.

The reason I submitted this review paper is because the title of the special issue is 'Chronic Rhinosinusitis: Aetiology, Immunology and Treatment', and I don't think there will be any special issues.

Reviewer 3 Report (New Reviewer)

please add a paragraph about on the management of eosinophilia and post-biological hypereosinophilia;

Please discuss the correlation between eosinophils and polyps doi: 10.3390/jpm12081215.

Please discuss potential biomarker of tissue eosinophils pro and cos

Author Response

I thank the editors and referees of the ‘International Journal of Molecular Sciences’ for taking their time to review my article.

I have made some corrections and changed some parts in the manuscript after going over the referee’s comments.

please add a paragraph about on the management of eosinophilia and post-biological hypereosinophilia;

Answer) Biologics associated eosinophilia was mentioned at line 406-409. As ‘However, treatment with dupilumab can cause transient blood eosinophilia, which is re-lated to the inhibition of eosinophil migration to the tissue. Therefore, the selection of bio-logics should be personalized depends on the pathophysiology of CRS.’ And reference 88 was added.

Please discuss the correlation between eosinophils and polyps  /jpm12081215.

Answer) Relationship between eosinophils and CRSwNP was added at line 81-84 as ‘Elevated blood eosinophil counts correlated with tissue eosinophilic inflammation and to the disease severity and the recurrence of CRS after sinus surgery [22,23]. Eosinophil counts in peripheral blood might be a biomarker for severe intractable CRS with type 2 in-flammation.’ And Reference 22 & 23 were added.

Please discuss potential biomarker of tissue eosinophils pro and cos

Answer) To mention about the biomarker of tissue eosinophils, ‘These Th2 cytokines have been suggested as potential predictors of clinical severity and treatment outcome. Among several Th2 cytokines, tissue and nasal secretion IL-5 seems to be an important biomarker, which correlated with tissue eosinophil count and clinical severity of CRS [36,37].’ Was added at line 139-142. And refered 36 & 37 were added.

I hope the revised manuscript will better meet the requirements of the ‘International Journal of Molecular Sciences’ for publication.

Thank you.

Round 2

Reviewer 3 Report (New Reviewer)

Regarding the comment:..... please add a paragraph about on the management of eosinophilia and post-biological hypereosinophilia;

 Biologics associated eosinophilia was mentioned at line 406-409. As ‘However, treatment with dupilumab can cause transient blood eosinophilia, which is re-lated to the inhibition of eosinophil migration to the tissue. Therefore, the selection of bio-logics should be personalized depends on the pathophysiology of CRS.’ And reference 88 was added.....

The authors should discuss the management not the only the pathophysiology.

Author Response

I thank the editors and referees of the ‘International Journal of Molecular Sciences’ for taking their time to review my article.

I have made some corrections and changed some parts in the manuscript after going over the referee’s comments.

The authors should discuss the management not the only the pathophysiology.

Answer) To mention about the management of eosinophilia, 'Although most of the eosinophilia is transient and do not develop clinical symptoms, treatment with prednisolone or dual therapy with anti-IL-5/5R monoclonal antibody may be needed in some cases [89]. ' was added at line 408-410.
And Ref 89 was added. 

Thank you.

This manuscript is a resubmission of an earlier submission. The following is a list of the peer review reports and author responses from that submission.

Round 1

Reviewer 1 Report

Shin et al reviewed the immunopathological roles of eosinophils in ECRS. In general, the manuscript is well written. However, there is room for improvement. The information is summarized in tables, but this is not very useful to the reader and can be found in other papers and review articles. The source of information in Table 2 is unclear since it lacks references. It would be better to include figures to make this paper more useful. The presence of graphical abstract has become essential for recent review papers, I strongly recommend the authors include at-a-glance information.

Line 43; Type 2 immune response is associated with allergic disease and parasitic infection. If the context is supposed to be sinus, then the parasitic infection seems strange, so please clarify if the topic is generalized to immune response. In the introduction, the paragraphs are long and difficult to understand, so it is better to break up the topics.

Line 73; eosinophils do not produce a significant amount of IL-5.

Line 91; The contents of the first paragraph seem to be generally accepted by eosinophil researchers, but need citation (review might be appropriate).

Line 108-112, 157-159, 188-191; Citation(s) needed.

For CRS classification, EPOS should be discussed.

Fokkens, W. J., et al. (2020). "European Position Paper on Rhinosinusitis and Nasal Polyps 2020." Rhinology 58(Suppl S29): 1-464.

Fokkens, W. J., et al. (2020). "Executive summary of EPOS 2020 including integrated care pathways." Rhinology 58(2): 82-111.

Line 138; The effect of IL-4 and IL-13 on eosinophils is indirect.

Preliminary discussion on the part of Aspergillus and eosinophils.

The pathogenesis of fungal allergy has been comprehensively described in previous reviews. For instance,

Bartemes KR, Kita H. Innate and adaptive immune responses to fungi in the airway. The Journal of allergy and clinical immunology. 2018;142(2):353-363.

Dykewicz MS, Rodrigues JM, Slavin RG. Allergic fungal rhinosinusitis. The Journal of allergy and clinical immunology. 2018;142(2):341-351.

The topic of the immune response against fungi is too broad, the authors should focus on the direct interaction of the fungi with eosinophils. It should be precisely discussed. In line 278, the following paper did not say the antifungal properties of EETs. Instead, the loss of antifungal capacities might contribute to the pathogenesis of allergic fungal diseases.

Muniz VS, Silva JC, Braga YAV, et al. Eosinophils release extracellular DNA traps in response to Aspergillus fumigatus. The Journal of allergy and clinical immunology. 2018;141(2):571-585 e577.

Ueki S, Hebisawa A, Kitani M, Asano K, Neves JS. Allergic Bronchopulmonary Aspergillosis-A Luminal Hypereosinophilic Disease With Extracellular Trap Cell Death. Front Immunol. 2018;9:2346.

Line 261; Ref 66 did not study mitochondrial DNA. There is debates about the source (mitochondrial vs nuclear) and original cell status (dead or alive) of EETs.

Mukherjee, M., et al. (2018). "Eosinophil Extracellular Traps and Inflammatory Pathologies-Untangling the Web!" Front Immunol 9: 2763.

In general, the most abundant proteins in nuclear-derived extracellular traps are histones that have cytotoxic effects. Allam, R., et al. (2014). "Extracellular histones in tissue injury and inflammation." J Mol Med (Berl) 92(5): 465-472.

Silk, E., et al. (2017). "The role of extracellular histone in organ injury." Cell Death and Disease 8(5): e2812. Sollberger, G., et al. (2018). "Neutrophil Extracellular Traps: The Biology of Chromatin Externalization." Dev Cell 44(5): 542-553.

ECRS is frequently associated with ear diseases. There are several reports on EETs, For instance,

Ohta, N., et al. (2019). "Possible clinical implication of eosinophil extracellular traps in eosinophilic otitis media." Otorhinolaryngology-Head and Neck Surgery 4(4).
Ohta, N., et al. (2018). "ETosis-derived DNA trap production in middle ear effusion is a common feature of eosinophilic otitis media." Allergol Int 67(3): 414-416.

It is supposed to broaden the reader's interest and knowledge.

Author Response

I thank the editors and referees of the ‘International Journal of Molecular Sciences’ for taking their time to review my article.

I have made some corrections and changed some parts in the manuscript after going over the referee’s comments.

Shin et al reviewed the immunopathological roles of eosinophils in ECRS. In general, the manuscript is well written. However, there is room for improvement. The information is summarized in tables, but this is not very useful to the reader and can be found in other papers and review articles. The source of information in Table 2 is unclear since it lacks references. It would be better to include figures to make this paper more useful. The presence of graphical abstract has become essential for recent review papers, I strongly recommend the authors include at-a-glance information.

Answer) Table 2 has been changed to Figure 1.

Line 43; Type 2 immune response is associated with allergic disease and parasitic infection. If the context is supposed to be sinus, then the parasitic infection seems strange, so please clarify if the topic is generalized to immune response. In the introduction, the paragraphs are long and difficult to understand, so it is better to break up the topics.

Answer) Line 43 ‘It is associated with allergic diseases and responds to defense against parasitic infections.’ Part was deleted as recommended.

Line 73; eosinophils do not produce a significant amount of IL-5.

Answer) I have deleted the content you pointed out (Line 73–75) to reduce confusion for the reader.

Line 91; The contents of the first paragraph seem to be generally accepted by eosinophil researchers, but need citation (review might be appropriate).

Answer) We added reference No 18 and 22 to the first paragraph of session 2, as recommended.

Line 108-112, 157-159, 188-191; Citation(s) needed.

Answer) For line 108-112, reference 24 and 25 were added.
        For line 157-159, reference 41 was added.
        For line 188-191, reference 56 was added.

For CRS classification, EPOS should be discussed.

Fokkens, W. J., et al. (2020). "European Position Paper on Rhinosinusitis and Nasal Polyps 2020." Rhinology 58(Suppl S29): 1-464.

Fokkens, W. J., et al. (2020). "Executive summary of EPOS 2020 including integrated care pathways." Rhinology 58(2): 82-111.

Answer) In introduction, we mentioned CRS classification, based on the EPOS 2020 and reference 7 was changed as recommend.

And Line 36-38 was changed as ‘Based on the endotype dominance, CRS is divided into either type 2 or non-type 2 [6-8]. Non-type 2 CRS can be divided into type 1 and type 3.’

Line 138; The effect of IL-4 and IL-13 on eosinophils is indirect.

Answer) To clarify, ‘directly or indirectly’ was added as ‘Th2 cytokines, such as IL-4, IL-5, and IL-13, directly or indirectly promote eosinophilia by influencing ….’

Preliminary discussion on the part of Aspergillus and eosinophils.

The pathogenesis of fungal allergy has been comprehensively described in previous reviews. For instance,

Bartemes KR, Kita H. Innate and adaptive immune responses to fungi in the airway. The Journal of allergy and clinical immunology. 2018;142(2):353-363.

Dykewicz MS, Rodrigues JM, Slavin RG. Allergic fungal rhinosinusitis. The Journal of allergy and clinical immunology. 2018;142(2):341-351.

The topic of the immune response against fungi is too broad, the authors should focus on the direct interaction of the fungi with eosinophils. It should be precisely discussed.

Answer) To focus on the interaction between fungi and eosinophils, 1) line 221-224 & line 229-232 were deleted.

And the fungi and eosinophil interactions were described as ‘Fungi can directly activate or induce the release of granule proteins in several way, 1) interaction with G protein-couple receptor with calcium-dependent manner [62], 2) interaction of fungal β-glucan with β2 integrin molecule of cell membrane [63], 3) interaction of fungal aspartate protease with protease activated receptor-2 [64], and 4) interaction of fungal hyphae or conidia with the CD11b extracellular I-domain [65]. Fungi can also in-directly induce the accumulation of eosinophils through the enhanced production of IL-33 or cysteinyl leukotrienes in airway mucosa [66] These chemical mediators activate IL-5 and IL-13 producing group 2 innate lymphoid cells, and then induce the production, recruitment, and activation of eosinophils.’

In line 278, the following paper did not say the antifungal properties of EETs. Instead, the loss of antifungal capacities might contribute to the pathogenesis of allergic fungal diseases.

Muniz VS, Silva JC, Braga YAV, et al. Eosinophils release extracellular DNA traps in response to Aspergillus fumigatus. The Journal of allergy and clinical immunology. 2018;141(2):571-585 e577.

Ueki S, Hebisawa A, Kitani M, Asano K, Neves JS. Allergic Bronchopulmonary Aspergillosis-A Luminal Hypereosinophilic Disease With Extracellular Trap Cell Death. Front Immunol. 2018;9:2346.

Answer) According to the Muniz et al. study, they mentioned as ‘EETs lack the killing or fungistatic activities against A fumigatus. However, what may be the function of EETs in ABPA?’. However, in CRS, eosinophils move to outside of nasal mucosa and move to fungi and release histones and DNA to remove fungus. During that time, eosinophils release toxic granule proteins, which influence epithelial barrier function and aggravate inflammation and invasion of pathogens. So EET formation is closed associated with disease severity in CRS [79].

The role of EET in the pathogenesis of ABPA and CRS is likely to be slightly different.

REF.)

  1. Hwang, C.S.; Park, S.C.; Cho, H.J.; Park, D.J.; Yoon, J.H.; Kim, C.H. Eosinophil extracellular trap formation is closely associated with disease severity in chronic rhinosinusitis regardless of nasal polyp status. Sci Rep 2019, 9, 8061, doi:10.1038/s41598-019-44627-z.

Line 261; Ref 66 did not study mitochondrial DNA. There is debates about the source (mitochondrial vs nuclear) and original cell status (dead or alive) of EETs.

Mukherjee, M., et al. (2018). "Eosinophil Extracellular Traps and Inflammatory Pathologies-Untangling the Web!" Front Immunol 9: 2763.

Answer) Ref 66, ‘Ueki, S.; Melo, R.C.; Ghiran, I.; Spencer, L.A.; Dvorak, A.M.; Weller, P.F. Eosinophil extracellular DNA trap cell death mediates lytic release of free secretion-competent eosinophil granules in humans. Blood 2013, 121, 2074-2083, doi:10.1182/blood-2012-05-432088.’ was changed as ‘Yousefi, S.; Simon, D.; Simon, H.U. Eosinophil extracellular DNA traps: molecular mechanisms and potential roles in disease. Curr Opin Immunol 2012, 24, 736-739, doi:10.1016/j.coi.2012.08.010.’

In general, the most abundant proteins in nuclear-derived extracellular traps are histones that have cytotoxic effects. Allam, R., et al. (2014). "Extracellular histones in tissue injury and inflammation." J Mol Med (Berl) 92(5): 465-472.

Answer) We mentioned about that as ‘In general, the most abundant proteins in nuclear-derived extracellular traps are histones that have cytotoxic effects [77].’
77.       Allam, R.; Kumar, S.V.; Darisipudi, M.N.; Anders, H.J. Extracellular histones in tissue injury and inflammation. J Mol Med (Berl) 2014, 92, 465-472, doi:10.1007/s00109-014-1148-z.

Silk, E., et al. (2017). "The role of extracellular histone in organ injury." Cell Death and Disease 8(5): e2812. Sollberger, G., et al. (2018). "Neutrophil Extracellular Traps: The Biology of Chromatin Externalization." Dev Cell 44(5): 542-553.

ECRS is frequently associated with ear diseases. There are several reports on EETs, For instance,

Ohta, N., et al. (2019). "Possible clinical implication of eosinophil extracellular traps in eosinophilic otitis media." Otorhinolaryngology-Head and Neck Surgery 4(4).

Ohta, N., et al. (2018). "ETosis-derived DNA trap production in middle ear effusion is a common feature of eosinophilic otitis media." Allergol Int 67(3): 414-416.

Answer) Since this paper was written on the topic of eosinophils & CRS, EET and OME were not mentioned.

It is supposed to broaden the reader's interest and knowledge.

I hope the revised manuscript will better meet the requirements of the ‘International Journal of Molecular Sciences’ for publication.

Thank you.

Reviewer 2 Report

A very good article overall but lacks focus on the 'role' of eosinophils. the article should also ask to what extent do eosinophils drive disease.

1. The authors have set a broad title of 'CRS and eosinophils', so have set themselves a very challenging area to write upon. Also the novel part of the title is the 'role' of the eosinophil in CRS. The authors are not clear about explaining disease pathogenesis, maintenance and disease exacerbation in terms of eosinophil biology (epithelial injury, expanding mucus, immune amplification, remodeling, immune dysregulation-each according to eosinophil role such as MBP, IL-13, TGFB1, infection control or failure as examples). At present the article just describes eosinophil biology. 

2. My advice is to define CRS, then eCRS according to EPOS2020, then focus the article on nasal polyp disease subtype of eCRS. We do not have enough literature on other types of eCRS to discuss in any detail. I would only discuss T2 high inflammation and not mention Type 1 or 3 inflammation.

3. The authors describe the biology of eosinophils to a good standard, but fail to explain how eosinophils can drive eCRS in terms of disease symptoms (nasal obstruction, mucus, smell loss, systemic symptoms), and mention only inflammation but not the consequences such as remodeling of tissue in eCRS, and how relevant all this is to disease activity and propagation. 

4. The most helpful insight into the role of eosinophils come from studies of intervention in CRSwNP with eosinophilic depletion strategies using biologics as an example. Generally the results are NOT impressive, so makes one wonder exactly how important eosinophils are to eCRS. With olfaction, IL-13/4 blockade with Dupilumab rather than IL-5 block leads to smell improvement. Thus how important are eosinophils to olfaction in CRSwNP.

5. Thus in summary, I would focus on a particular eCRS subtype, explain how eosinophils may or may not cause disease , and use the biologics data to support potential hypotheses on mechanistic roles for eosinophils in CRS. 

Author Response

I thank the editors and referees of the ‘International Journal of Molecular Sciences’ for taking their time to review my article.

I have made some corrections and changed some parts in the manuscript after going over the referee’s comments.

  1. The authors have set a broad title of 'CRS and eosinophils', so have set themselves a very challenging area to write upon. Also the novel part of the title is the 'role' of the eosinophil in CRS. The authors are not clear about explaining disease pathogenesis, maintenance and disease exacerbation in terms of eosinophil biology (epithelial injury, expanding mucus, immune amplification, remodeling, immune dysregulation-each according to eosinophil role such as MBP, IL-13, TGFB1, infection control or failure as examples). At present the article just describes eosinophil biology. 

    Answer) As pointed, most of the manuscript is focused on the biologic role of eosinophils, so the title has been changes as ‘The biologic role of eosinophils in development of eosinophilic chronic rhinosinusitis’

  1. My advice is to define CRS, then eCRS according to EPOS2020, then focus the article on nasal polyp disease subtype of eCRS. We do not have enough literature on other types of eCRS to discuss in any detail. I would only discuss T2 high inflammation and not mention Type 1 or 3 inflammation.

Answer) In introduction, we mentioned CRS classification, based on the EPOS 2020 and reference 7 was changed as recommend.

And Line 36-38 was changed as ‘Based on the endotype dominance, CRS is divided into either type 2 or non-type 2 [6-8]. Non-type 2 CRS can be divided into type 1 and type 3.’

  1. The authors describe the biology of eosinophils to a good standard, but fail to explain how eosinophils can drive eCRS in terms of disease symptoms (nasal obstruction, mucus, smell loss, systemic symptoms), and mention only inflammation but not the consequences such as remodeling of tissue in eCRS, and how relevant all this is to disease activity and propagation. 

Answer) Title of session 3 was changed as ‘Immunopathologic role of eosinophils in CRS’ and mentioned about the effects of eosinophils on not only olfaction but also mucus production and tissue remodeling.

  1. The most helpful insight into the role of eosinophils come from studies of intervention in CRSwNP with eosinophilic depletion strategies using biologics as an example. Generally the results are NOT impressive, so makes one wonder exactly how important eosinophils are to eCRS. With olfaction, IL-13/4 blockade with Dupilumab rather than IL-5 block leads to smell improvement. Thus how important are eosinophils to olfaction in CRSwNP.

Answer) Recently, biologics targeting eosinophils are important therapeutic strategies for severe intractable CRS. Because, this paper is mainly focused on the pathophysiologic role of eosinophils, we briefly mentioned about biologics at the end of the Summary as ‘ For the treatment of severe, uncontrollable ECRS, several biologics targeting the underly-ing type 2 inflammation are developed and approved, such as dupilumab, omalizumab, and mepolizumab [85]. Dupilumab (anti-IL-4Rα monoclonal antibody) and mepolizumab (anti-IL-5 monoclonal antibody) more directly affect eosinophils. Although, omalizumab (anti-IgE monoclonal antibody) does not directly influence eosinophils, there is a significant correlation between levels of total IgE and levels of IL-5, ECP, and number of eosinophils [34]. These biologicals can improve sense of smell, quality of life, nasal polyp scores, and CT score, and reducing the need for rescue surgery with somewhat different efficacy and safety [86]. If can gain precise knowledge about the immunopathologic role of eosinophils in the pathogenesis of CRS, controlling eosinophil maturation, recruitment, survival, and activation could be effective therapeutic strategies.’

  1. Thus in summary, I would focus on a particular eCRS subtype, explain how eosinophils may or may not cause disease , and use the biologics data to support potential hypotheses on mechanistic roles for eosinophils in CRS. 

Answer) Recently published review paper mentioned role of eosinophils and biologics, so I did not describe about that in detail.

Bochner BS, Stevens WW. Biology and function of eosinophils in chronic rhinosinusitis with or without nasal polyps. Allergy Asthma Immunol Res 2021; 13(1):8-22.

I hope the revised manuscript will better meet the requirements of the ‘International Journal of Molecular Sciences’ for publication.

Thank you.

Round 2

Reviewer 1 Report

The manuscript has been revised, although I found it a little difficult to decide what to make of this work. The title has been changed, but my concern is that this review does not really present “the biological role of eosinophils in development of ECRS”. The authors should have more citations and discuss the biological roles fully.

Figure 1;

Line 291;

These stimuli have been shown to induce cytolytic EET release from the nuclear origin (Ueki et al. Blood. 2013).

Author Response

I thank the Reviewers of the ‘International Journal of Molecular Sciences’ for taking their time to review my article.

I have made some corrections and changed in the manuscript after going over the comments.

The manuscript has been revised, although I found it a little difficult to decide what to make of this work. The title has been changed, but my concern is that this review does not really present “the biological role of eosinophils in development of ECRS”. The authors should have more citations and discuss the biological roles fully.

Answer) To reduce readers confusion, the title of the article has been changed as ' Immunopathologic role of eosinophils in eosinophilic chronic 2 rhinosinusitis' and subtitle of session 3 was changed as 'Pathologic role of eosinophils in CRS'

Figure 1;

Answer) Fig 1 was change as Figure 1.

Line 291;

These stimuli have been shown to induce cytolytic EET release from the nuclear origin (Ueki et al. Blood. 2013).

Answer) Reference 74 was changed as recommended. 

Thank you.

Round 3

Reviewer 1 Report

Line 291;These stimuli have been shown to induce cytolytic EET release from the nuclear origin, not mitocondrial.